



# Contrasting dynamics of lake- and marine-terminating glaciers under same climatic conditions

Florian Vacek[1], Faezeh M. Nick[1], Douglas Benn[2], Maarten P.A. Zwarts[3], Walter Immerzeel[1], and Roderik S. W. van de Wal[1, 4, 5]

[1]Department of Physical Geography, Utrecht University, Utrecht, The Netherlands
[2]School of Geography and Sustainable Development, University of St Andrews, St Andrews, UK
[3]Earth Simulation Laboratory, Utrecht University, Utrecht, The Netherlands
[4]Institute for Marine and Atmospheric research Utrecht, Utrecht University, Utrecht, The Netherlands
[5]Royal Netherlands Meteorological Institute (KNMI), De Bilt, The Netherlands

**Correspondence:** Florian Vacek (f.vacek@uu.nl)

**Abstract.** In Greenland, mass wasting through frontal ablation occurs not only at the ice-ocean interface but also at the ice-lake intersection. Recent studies have found that lakes cover 10% of the entire ice sheet margin and stress the importance of understanding frontal dynamics in lacustrine settings. However, relatively little is known about how lake-terminating glaciers compare to marine-terminating glaciers under the same climatic conditions. At a unique study site in South Greenland, a lake and a marine terminus are part of the same glacier system, subject to the same climatic forcings and fed by the same upstream ice masses. In this study, we analyse the drivers of change at both glacier fronts and compare their dynamics with a comprehensive remote sensing dataset supported by climate and ocean model output. Furthermore, during two field campaigns, we collected lake bathymetry data alongside temperature and lake level measurements. We find that despite being subject to the same climatic forcing and fed by the same upstream ice masses, the two termini show contrasting front dynamics in the long- and short-term. We argue that low subaqueous melt rates in the lake are the main driver of these differences. Furthermore, we find lake ice to limit calving activity, similar to an ice mélange at marine-terminating glaciers. A massive disintegration event of more than 3 km of the lake terminus showcases the possibility of rapid mass loss at lake-terminating glaciers in Greenland. Our results stress that lake- and marine-terminating glaciers require different parameterisations when included in model simulations of the Greenland Ice Sheet.

## 1 Introduction

The Greenland Ice Sheet (GrIS) is one of the largest contributors to contemporary sea level rise (Fox-Kemper et al., 2021). Its mass loss is partitioned by surface melt and dynamic ice discharge into the ocean through calving and submarine melt (Mouginot et al., 2019; Otosaka et al., 2023). Both modes of mass loss have increased strongly in the 21st century (Shepherd et al., 2020) as a consequence of higher air temperatures (Hanna et al., 2012) and warm ocean water (Straneo and Heimbach, 2013; Wood et al., 2021). Future projections of mass loss from Greenland vary strongly between different models and climate scenarios (Goelzer et al., 2020), with one of the largest uncertainty arising from the difficulty in resolving the frontal dynamics



of tidewater glaciers. At the ice-ocean boundary, ice is rapidly lost through iceberg calving and submarine melt (Benn et al., 2007; Truffer and Motyka, 2016), which can alter the force balance and consequently trigger a speed up of glacier flow, resulting in a higher dynamic mass loss (Joughin et al., 2008; Nick et al., 2009). Calving and submarine melt are complex processes that
are strongly influenced by the characteristics of the terminal environment, such as ocean temperature (Wood et al., 2021), tides (Holmes et al., 2023), wave action (Pętlicki et al., 2015), plume formation (Cook et al., 2021), and buttressing through an ice mélange (Barnett et al., 2023; Wehrlé et al., 2023).

These complex interactions of ice and water are not unique to marine settings, but also occur in lacustrine settings, potentially further increasing dynamic ice loss in Greenland. Recent studies have identified a large number of ice-marginal lakes with more
than 2300 occurrences larger than 0.05 km$^2$(How et al., 2021). Together, these lakes constitute 10% of the total GrIS margin (up to 26% regionally) and accelerate ice flow velocities and margin retreat, compared to the land-terminating sections of the ice sheet (Mallalieu et al., 2021; Carrivick et al., 2022). Furthermore, the number of ice-marginal lakes and their impact on the GrIS is expected to grow as the ice sheet recession continues. Carrivick et al. (2022) identified thousands of over-deepenings that, upon retreat, could transform into ice-marginal lakes. A study of central and southwestern Greenland showed a notable
44 % increase in lakes between 1987 and 2010 (Carrivick and Quincey, 2014).

Lake-terminating glaciers are generally known to have lower flow velocities and ice discharge compared to marine-terminating glaciers in Greenland (Mallalieu et al., 2021; Carrivick et al., 2022). However, it is unclear which processes control their dynamics at the terminus in detail. Furthermore, a comparison of the two different terminal environments under the same climatic conditions is lacking.

In South Greenland, a unique study site poses the possibility of conducting such a comparison. At Qooqqup Sermia, both a marine and lake terminus are part of the same glacier system, subject to the same climatic forcings, fed by the same upstream ice masses and also comparable in size. Despite its uniqueness, research on the glacier system is limited to early descriptions from Weidick (1959, 1963) and Warren and Glasser (1992). Weidick summarises earlier expeditions to the region and notes that no reliable glacier positions can be inferred from their reports. Almost thirty years later, Warren and Glasser describe a
non-linear response of the glacier termini to climate warming. In addition, there are paleoglaciological studies mapping and dating late Holocene moraines of the Qooqqup Sermia system (maximum Holocene extent about 1.5 ka BP) (Bennike and Sparrenbom, 2007; Winsor et al., 2014) and a reconstruction of paleo ice thickness (Puleo and Axford, 2023).

This study aims to compare glacier dynamics at the marine- and lake-terminating glacier and to discuss drivers of differences between the two environments. To achieve this, we used a combination of field observations, remote sensing, and modelled
ocean and climate data. We compiled a dataset consisting of margin outlines, surface velocities, elevation changes, lake ice cover, surface runoff, subsurface ocean temperature, lake temperature and lake bathymetry. To put our findings into context, we also describe the general characteristics and historical evolution of Qooqqup Sermia. Our findings suggest that lake- and marine-terminating glaciers need to be included separately in models simulating the evolution of the GrIS.



## 2 A unique study site in South Greenland

The Qooqqup Sermia system is located close to the town of Narsarsuaq in South Greenland (Fig. 1). The glacier system drains a catchment of approximately 2400 km$^2$ (Mouginot and Rignot, 2019) and has three major outlet glaciers: Kiattuut Sermiat terminating on land, Qooqqup Sermia terminating in the Qooroq Fjord, and an unnamed glacier, which splits from Qooqqup Sermia and flows into Lake Motzfeldt. In the following, we refer to the tidewater glacier as the *Marine Terminus* (MT) and to the glacier flowing into Lake Motzfeldt as the *Lake Terminus* (LT).

The marine terminus is about 1.7 km wide and the glacier front reaches heights up to 65 m above water (based on UAV field surveys in August 2024). The water depth directly at the glacier front is unknown. However, at approximately 3 km from the current glacier front, the fjord has a depth of 350–375 m (OMG, 2019). With increasing distance from the glacier, the fjord deepens to ∼ 480 m before becoming more shallow again toward the fjord mouth. The fjord mouth is characterised by a large Holocene submarine moraine (less than 50 m water depth), which prevents larger icebergs from exiting into the adjacent
Tunulliarfik fjord (Vorndran and Sommerhoff, 1974; Weidick, 1963).

The lake terminus is currently about 1.6 km wide and has a glacier front height of up to 30 m above lake level (based on UAV field surveys in August 2024). Lake Motzfeldt is approximately 15 km long and 2 km wide and lies at an elevation of 160 m above sea level. Besides the lake terminus of Qooqqup Sermia, a second, unnamed glacier flows into the lake on the eastern side. Throughout the year, the lake is covered with several large tabular icebergs (> 0.1 km$^2$) and many smaller icebergs
(visible in Fig. 2).

The closest weather station to MT and LT is located at the Narsarsuaq airport. At this station, the mean annual air temperature (MAAT) between 1981 and 2010 was 1.1 °C (Drost Jensen, 2025). During the last 15 years (2010–2024) the MAAT was more than 1 °C higher with an average of 2.2 °C. Mean monthly temperatures are lowest in February (-7.4 °C ) and highest in July (10.8 °C).

## 75 3 Methods

To understand the dynamics and drivers of change at the marine and the lacustrine glacier margin, we used a combination of remote sensing, field methods, and analysis of climate and ocean model data. Specifically, we used remote sensing data to outline glacier front positions, measure surface elevation changes, ice flow velocities and to identify lake ice duration. In the field, we conducted a bathymetric survey of Lake Motzfeldt alongside temperature and lake level measurements during two
field campaigns in 2024 and 2025. Furthermore, we used the output of the regional climate model RACMO and the Ocean Reanalysis System ORAS5.

### 3.1 Glacier front positions

We manually outlined glacier front positions for the marine- and the lake-terminating glacier based on optical satellite images from Landsat 4, 5, 7 and 8 for the period 1992–2024. The images were downloaded using Google Earth Engine, and the margins







**Figure 1.** Map of the study area. The blue (MT) and red (LT) lines indicate the elevation profiles used in this study. The grey squares indicate locations for which we show ice surface velocity. Yellow circles indicate CTD measurements with the respective number. The yellow pentagon shows the location of the temperature-depth sensor. The hillshade is based on the ArcticDEM (Porter et al., 2023).

were digitised using ArcGIS Pro. For Landsat 4 and 5, we used the near-infrared band (band 4) at a spatial resolution of 30 m. For Landsat 7 and 8, we used the panchromatic band (band 8), with a spatial resolution of 15 m. We identified additional front positions for 1987 based on orthorectified 2 m aerial images (Korsgaard et al., 2016) and for 1953 based on aerial images from The Danish Agency of Climate Data (see supplements for image details). The 1953 images were manually georeferenced based on tie points. We outlined a total of 389 and 308 front positions for MT and LT, respectively. While Landsat 4, 5 and 7



images only allowed for a relatively sparse temporal resolution with about 3–5 images per year, cloud-free Landsat 8 images were available on average every 15 (MT) and 17 days (LT), allowing for a seasonal analysis over the 11 years from 2014 to 2024. We quantified relative changes in the terminus positions using the variable box method as implemented in the *Margin change Quantification Tool* (MaQiT) (Lea, 2018). By considering the entire width of the glacier terminus, the method allows the detection of uneven changes along the glacier front. For the lake terminus, we identify major calving events by extracting

glacier front position changes larger than 50 m between two images. Since the number of calving events is relatively small, we visually verify all instances based on satellite images.

## 3.2 Ice surface elevation change

We evaluated changes in ice surface elevations with two datasets. First, a 25 m resolution historical DEM (Korsgaard et al., 2016). This DEM was constructed with stereo-photogrammetry based on $\sim 2$ m aerial images covering the entire Greenland

margin. The Danish Agency for Data Supply and Efficiency (SDFE) acquired the images in aerial campaigns between 1978 and 1987. In our region of interest, the underlying images of the DEM were taken in 1987. The DEM is co-registered to ICESat laser altimetry elevations, and the authors report an accuracy better than 10 m horizontally and 6 m vertically (Korsgaard et al., 2016).

Second, we used the ArcticDEM strip collection, version 4.1 (Porter et al., 2022). The dataset is a compilation of 2 m

resolution DEMs constructed with stereo photogrammetry based on submeter resolution Maxar satellite images (Noh and Howat, 2015). DEMs are available for single stereo image pairs between 2007 and 2023 (2012–2023 in our region). The DEMs vary in size and often do not cover both glacier termini at the same time. We accessed the DEMs via the FRIDGE portal of the Polar Geospatial Center (PGC, https://fridge.pgc.umn.edu/). To reduce the influence of snow cover on the measured elevation changes we filtered the dataset to exclusively contain DEMs acquired in July, August, or September. Following that,

we selected the first and the latest available DEM, which cover both glacier termini. The selected DEMs are from 14 August 2012 and 30 September 2023.

Porter et al. (2022) report an accuracy of the ArcticDEMs of about 4 m horizontally and vertically. To reduce the uncertainty in calculated elevation changes, we co-registered the DEMs with a method based on (Nuth and Kääb, 2011), as it is implemented in the Python package xDEM (xDEM contributors, 2024). For the co-registration, we masked out all glaciated areas

and water bodies and only used assumed stable terrain (bedrock areas). Additionally, we applied the reliability mask, which is provided with the ArcticDEM, to remove photogrammetric artefacts. With the co-registration, we were able to reduce the median elevation change above stable terrain from -3.12 to 0.001 m for the ArcticDEM and from -3.17 to 0.002 m for the historical DEM. The normalised median absolute deviation (NMAD), which gives an indication of dispersion, was reduced from 0.92 to 0.51 m and from 3.89 to 3.55 m for the ArcticDEM and historical DEM, respectively. Based on that and in line

with previous studies using a similar approach (Błaszczyk et al., 2019; Holt et al., 2024), we conclude that the ArcticDEM allows for the detection of changes less than 0.1 m yr$^{-1}$ for the observation period 2012–2023. Due to the long time interval, we are also confident in detecting changes $\geq 0.1$ m yr$^{-1}$ between the historical DEM from 1987 and the ArcticDEM.





To avoid point sampling of local features such as crevasses, we resampled the co-registered ArcticDEMs to 25 m resolution using bilinear interpolation. Furthermore, we extracted width-averaged values of a 400 m wide band following the glacier centre lines (Fig. 1).

### 3.3 Ice surface velocities

We extracted NASA ITS_LIVE ice surface velocities (Gardner et al., 2025) at both glacier fronts and about 8 km upstream of the marine terminus (locations indicated in Fig. 1). The ITS_LIVE dataset contains image pair velocities from Landsat 4, 5, 7, 8, 9, Sentinel 1 and 2. The velocities are constructed with the auto-RIFT feature tracking algorithm (Gardner et al., 2018; Lei et al., 2021). With the launch of Sentinel 1 and 2 satellites between 2014 and 2017 the amount of available image pair velocities has increased substantially. Therefore, we constrained our analysis to the period 2016–2024 with several thousand available image pair velocities.

The difficulty in analysing ITS_LIVE data lies in the different temporal base lines of image pairs, making them not comparable to each other. While some image pairs might be separated by only a few days, others might be separated by more than a year. To address this issue, we make use of the Python package TICOI (Temporal Inversion using Combination of Observations and Interpolation) (Charrier et al., 2025). The algorithm performs a temporal inversion to entangle the contribution of each image pair to the velocity at a given time. Furthermore, TICOI performs an interpolation in order to provide velocities at a regular, comparable interval.

The ITS_LIVE data is stored in cloud-optimised Zarr data cubes, from where we access the data directly through the TICOI Python package. Before running TICOI, we apply several filters to remove outliers. First, we filter out image pairs that have a flow direction differing by more than 45 ° from the median flow direction of all observations. Second, we filter out values below or above 80 % of the mean velocity. Finally, we filter out image pair velocities that have an error > 100 m in the error estimates provided by ITS_LIVE. We find that image pairs acquired by SAR sensors show a very large scatter for the slower flowing lake terminus. Therefore, we exclude Sentinel-1 image pairs for this glacier.

### 3.4 Temperature and lake level measurements

We measured three lake temperature profiles with an RBR concerto CTD measuring temperature, pressure and conductivity at 8 Hz. The manufacturer states an initial accuracy of $\pm 0.002$ °C for the temperature. Moreover, we measured a temperature and lake level time series for almost an entire year with an RBR Duet3. The sensor was placed inside a metal pipe and attached to a rock wall with Dyneema rope. The sensor was placed at approximately 5.4 m depth and measured temperature and pressure at a 30 minute interval between 09 August 2024 and 01 August 2025. Measurements were compensated for atmospheric pressure with a barometric unit placed close to the lake during the measurement period. The locations of all measurements are indicated in Fig. 1.



### 3.5 Bathymetric survey

We conducted a bathymetric survey of Lake Motzfeldt with a Kongsberg EA440 single-beam echo-sounder and a 38|200 khz
combi transducer. The instrument was mounted to a self-built aluminium frame attached to a 4.5 m long inflatable boat. An
Emlid Reach RS3 GNSS receiver was used for location input at 5 hz and was mounted on top of the transducer. During survey-
ing, the boat was manoeuvred at approximately 9 km h$^{-1}$ aiming to follow a survey grid with a spacing of 200 m. However,
large amounts of icebergs prevented us from doing so on several occasions, leading to areas with slightly more or slightly less
spacing between survey lines. The measurements were corrected for sound velocity based on the CTD measurements. The data
was manually cleaned of outliers in ArcGIS Pro. To produce a bathymetric map, we used the Python implementation of the
Generic Mapping Tool (PyGMT) (Tian et al., 2025). We first calculated a block-median with a grid cell-size of 25 m before
applying a spline interpolation function with a tension factor of 0.6. Finally, a grid was exported at 25 m resolution.

### 3.6 Floatation index

For the lake terminus we estimated the glaciers' floatation potential based on the ice thickness (assuming floatation) and the lake
depth. Following Archimedes' principle of buoyancy we calculated the ice thickness below water ($H_{bw}$) from the freeboard
($h$), an ice density ($\rho_i$) of 917 kg m$^{-3}$ and a water density ($\rho_w$) of 1000 kg m$^{-3}$ as:

$$H_{bw} = (\rho_w/(\rho_w - \rho_i) - 1)h \tag{1}$$

We derive the freeboard for the 1987 and 2012 DEMs by subtracting the respective lake levels from the glacier surface
elevation, where an overlap with the bathymetry data was present. Consequently, we calculate the floatation index ($I$) as:

$$I = H_{bw} - D \tag{2}$$

where $D$ is the lake depth. The glacier is floating where $H_{bw}$ is smaller than $D$ and grounded where $H_{bw}$ exceeds $D$. The
observed lake level changes introduce an error to $D$, which, however, is relatively small (< 2 m).

### 3.7 Lake ice

After initial tests to automatically detect lake ice based on SAR or optical satellite data, we find it unsuitable for Lake Motzfeldt
because of the immense amount of icebergs permanently covering the lake. Therefore, we fall back to a manual detection based
on a combination of optical satellite images and air temperature. We use GEEDiT (Lea, 2018) to manually label Sentinel-2,
Landsat-7, 8 and 9 images with *'full ice cover'*, *'partial ice cover'* and *'no ice cover'*. Based on long-term climatic means
from the Narsarsuaq weather station, we solely inspect images after mid-April (lake ice breakup) and after the first of October
(lake freeze up). Once the lake ice cover is established in the fall, we assume that it persists until breakup in May or June.
We verified this by checking all instances where daily mean temperatures rose above zero for more than 5 consecutive days.





Despite hindering automatic detection, the movement of the icebergs is an excellent indicator of the presence of lake ice in the manual classification. Without lake ice, icebergs move due to wind and water currents. However, as soon as the lake ice cover is established, the icebergs are locked in place. To identify lake ice, we also used the visual colour difference between ice and open water, cracks in the lake ice, and snow cover. Since a partial lake ice cover likely only has a minor buttressing effect, we

report the lake ice duration as the time between the first and last image labelled with *'full ice cover'* for every winter season.

### 3.8    Climate, runoff and ocean data

We acquire mean daily air temperature data for Narsarsuaq from the Danish Meteorological Institute (DMI) (Drost Jensen, 2025). The station with the ID 04270 is located at Narsarsuaq Airport at an elevation of approximately 34 m above sea level and approximately 12 km west of the marine terminus and 18 km west of the lake terminus (Fig. 1). The available air temperature

data spans from 1961 to present.

     We obtained freshwater runoff for Qooqqup Sermia from the Regional Atmospheric Climate Model (RACMO2.3p2), statistically downscaled to 1 km (Noël et al., 2018). We spatially average daily runoff data for the entire Qooqqup Sermia based on the catchment from (Mouginot and Rignot, 2019).

     We obtained subsurface ocean temperature data from the Ocean Reanalysis System 5 (ORAS5) (Zuo et al., 2019). We

extracted a time series with monthly resolution of the closest raster cell to the Tunulliarfik fjord mouth, which contains data up to at least 100 m deep. For this cell (60.8162 °N, -46.3505 °E) we depth average the potential temperature between 20 and 100 m depth.

### 4    Results

### 4.1    Historical front positions and elevation change

In the period analysed between 1953 and 2024, the two glacier termini show clear differences in their overall evolution of the terminus position, as well as in their retreat and advance patterns (Fig. 2). At first, both glacier fronts show a net advance between the first aerial observation in 1953 and the first satellite observations in 1992. Subsequently, however, they develop differently. The marine terminus reached its most advanced position in 1994, from where it gradually retreated about 1.3 km until 2004. Since 2004, the terminus has remained in a consistent position, with a seasonal advance and retreat pattern (Fig.

2a, blue shading). The lake terminus, on the other hand, held a stable position until 2012, when it abruptly retreated about 3.2 km within one year. Since this event, the glacier front has transitioned into a mode of long phases of gradual front advance followed by abrupt, large calving events (Fig. 2a, red shading), producing tabular icebergs.

     Based on a comparison of digital elevation models, we observe pronounced thinning at both glacier termini (Fig. 3). At the marine-terminating glacier, the highest thinning rates are observed close to the glacier front, with a rate of 2.2 m yr$^{-1}$, or a

total of 80 m, between 1987 and 2023. With increasing elevation, the thinning rates decrease continuously to approximately 1.7 m yr$^{-1}$ 8 km from the glacier front. At the lake terminus, the lowest thinning rates are observed across the floating glacier





**Figure 2.** Terminus positions of the marine terminus (left, blue) and the lake terminus (right, red). Panel (a) shows the earliest observations in 1953, the maximum positions in 1994 (MT) and 2012 (LT), and the areas of advance and retreat (blue and red shaded areas). The shading is bound by the local maximum and minimum positions after the respective retreat in 2004 and 2012. Background: Sentinel-2 image from 19 September 2024. Panel (b) shows the terminus retreat relative to the first satellite observations in 1992. The dotted boxes indicate the period for the shading in panel a. Years with available digital elevation models are marked above the x-axis.

tongue, only 0.3 m yr$^{-1}$ or 7.7 m in total before its disintegration in 2012. These thinning rates amount to approximately 10 % of the thinning above grounded areas, due to to buoyant adjustment. At LT, the highest thinning rates are observed upstream of what became the new glacier front after the disintegration in 2012, with a rate of 2.3 m yr$^{-1}$ or a total of 82 m (1987–2023).

By comparing thinning rates for both glaciers at the same elevation (> 200 m above sea level), we find that between 1987 and 2012, the thinning rates are similar, with 2.1 m yr$^{-1}$ (LT) and 2.0 m yr$^{-1}$ (MT). However, between 2012 and 2023, the



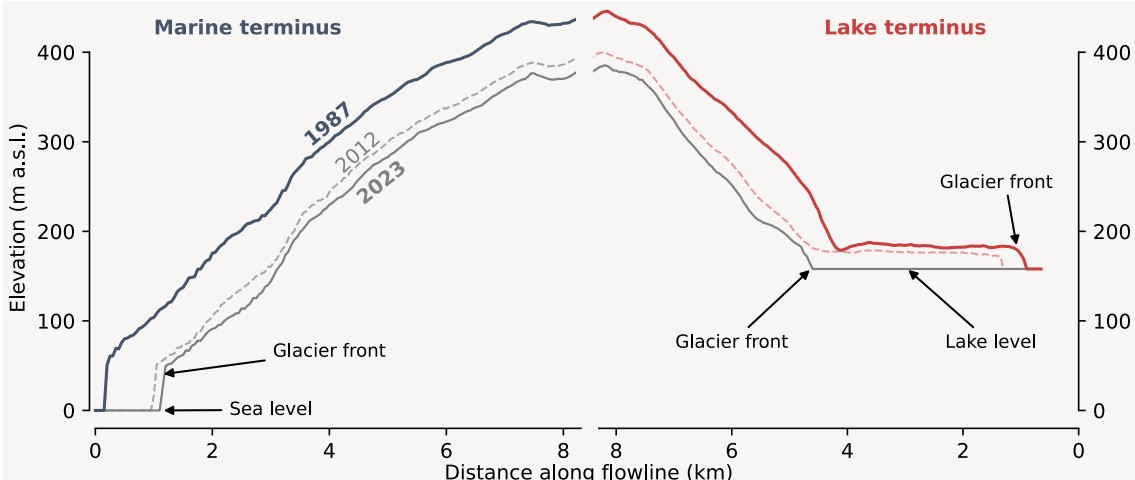

**Figure 3.** Surface elevation profiles for both glacier termini along the center lines indicated in Fig. 1. Glacier surface elevations are extracted from DEMs from 1987 (Korsgaard et al., 2016), 2012 and 2023 (Porter et al., 2022).

thinning rate of the lake-terminating glacier exceeded that of the marine-terminating glacier by 0.4 m yr$^{-1}$, with rates of 1.7 and 1.3 m yr$^{-1}$, respectively. Both glaciers clearly show lower thinning rates after 2012.

### 4.2 Bathymetry and floatation

The bathymetric survey of Lake Motzfeldt reveals an exceptionally deep lake with a maximum lake depth of 368 m (Fig. 4a, red dot) and an average of 131 m. The lake is generally shallower south of the lake bend and gradually deepens toward the north. The lake morphology is characterised by steep side walls and a flat, deep middle part. While no distinct moraine can be seen where the glacier was stagnant until 2012, a clearly visible ridge extends from the western lake shore to the middle of the lake (Fig. 4a). However, the composition of this ridge is unknown.

The floatation index shows that in 1987, extended parts of the glacier front were grounded on and upstream of the ridge, as well as on both lateral margins (Fig. 4b). On the ridge, the floatation index reaches values of up to 200 m, which means that the theoretical ice thickness, assuming floatation, exceeds the lake depth at that position by 200 m. Due to substantial thinning (see the previous section), most of the pixels classified as grounded in 1987 are classified as floating in 2012. The grounded area on the ridge was largely diminished with a maximum floatation index value of 80 in 2012. The DEM from 2012 shows

that, at that time, the lateral margins of the glacier had crumbled apart, leading to the loss of grounded areas on the sides.

### 4.3 Lake temperature and lake level

The three CTD measurements show a cold lake with a depth average of about 0.65, 0.68 and 0.71 °C for CTD 1, 2 and 3 respectively (Fig. A2). All CTD casts are coldest at the top and gradually become warmer toward the bottom. The coldest temperatures are found closest to the glacier front (CTD 1), and the warmest are farthest away (CTD 3). CTD 1 and 2 remain



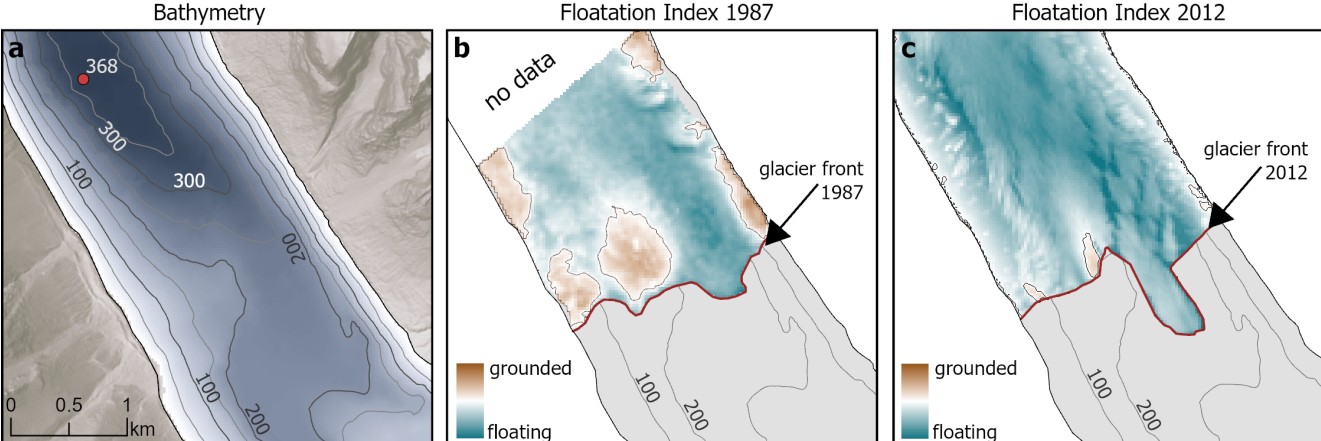

**Figure 4.** The bathymetry and floatation index at the LT glacier front. Panel (a) shows the results from the bathymetric survey of Lake Motzfeldt. The red dot marks the deepest point in the lake. Panel (b) and (c) show the floatation index at the glacier front in 1987 and 2012.

below 0.75 °C at all depths, while CTD 3 shows slightly warmer water at the bottom, with a maximum temperature of 0.86 °C. The time series of the temperature sensor that was placed from 2024 to 2025, shows that temperatures decreased from about 0.7 °C at deployment in August to 0.2 °C in November (Fig. A1). Between November and mid May, the lake water stays below 0.2 °C, with very little daily variation. Following that, temperatures rise again to a maximum of 0.8 °C in the end of July before starting to decrease again in September. The lake level, inferred from the sensor depth, remains approximately constant

between the beginning of October and the beginning of July. In July, the lake level rises by 1.6 m compared to the winter low. In August and September, the lake level is generally high (1 m above winter low); however, it shortly lowers by 0.6 m at the End of August.

## 4.4 Dynamics at the lake terminus

We tracked changes in glacier front position with high temporal resolution between 2014 and 2024 and found a characteristic

ice-shelf calving style at the lake-terminating glacier (Fig. 5). Following the disintegration of the ice tongue in 2012–2013, the LT glacier front has experienced prolonged phases of gradual front advance, interrupted by abrupt, large calving events. In some years, such as 2020, the advance phase persisted through the summer, with the glacier front advancing approximately 500 m over the entire year without any noticeable calving events. As shown in Fig. 2, the glacier advances primarily in the centre of the lake, with no lateral contact with the lake shore.

Between 2014 and 2024, we identified 15 major calving events, each resulting in a front retreat exceeding 50 m (Fig. 5). The largest single-event retreat measured 923 m, while the most substantial cumulative retreat measured about 1.4 km in the summer of 2017 during three consecutive events. From satellite images, we observed that these events produced large tabular icebergs, indicating floatation. All calving events coincided with the runoff season and occurred during ice-free conditions on Lake Motzfeldt (Figs. 5, 6).





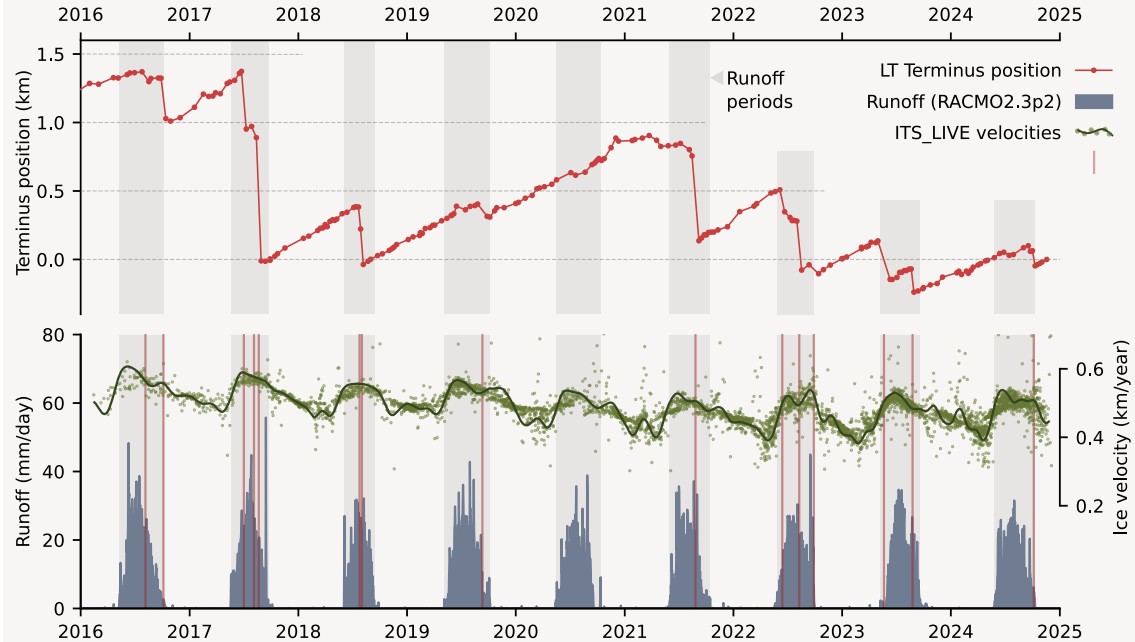

**Figure 5.** Dynamics at the lake terminus. The upper part of the figure shows the terminus position relative to the last observation. The lower part of the figure shows ice flow velocities at the glacier front from ITS_LIVE and TICOI (Gardner et al., 2025; Charrier et al., 2025). Daily runoff from RACMO2.3p2 (Noël et al., 2018) is shown with blue bars. Calving events, extracted from the front positions, are marked with red vertical lines in the lower part of the figure.

The duration of the full lake ice cover in Lake Motzfeldt averaged 212 days $yr^{-1}$, ranging from 187 days in 2015 to 230 days in 2021 (Fig. 6). Lake ice cover generally developed in October, with one exception in 2015 where it developed in early November. The earliest onset of the full ice cover was observed on 8 October 2022. Lake ice break up usually occurs in May or June, the earliest recorded on 2 May 2021 and the latest on 21 June 2015.

In the observed period 2016–2024, the ice flow velocities at the lake terminus ranged between 380 and 610 m $yr^{-1}$, with a

gradual decline in the average annual velocity from $\sim$ 544 m $yr^{-1}$ in 2016 to $\sim$ 472 m $yr^{-1}$ in 2024. A clear seasonal signal can be identified with a 15% increase from winter (DJF) to summer (JJA) velocities. Ice flow velocities usually increase with the beginning of the runoff season and drop again at the end of the runoff season. Calving events do not appear to have an effect on the glaciers' flow velocity (Fig. 5).

### 4.5 Dynamics at the marine terminus

Between 2014 and 2024, the marine-terminating glacier showed a distinct seasonal cycle, with front advance occurring on average between October and May and retreat from June through September (Fig. 7 and 8). Notable exceptions are the years 2019, 2020 and 2021, where retreat already started in February, April, and March, respectively. The average magnitude of front




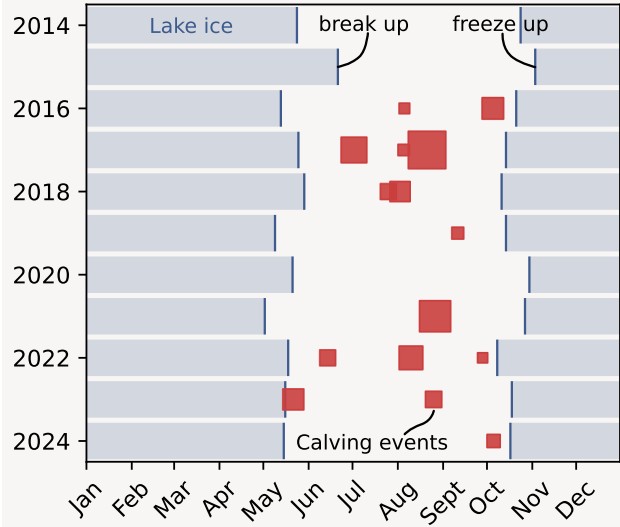

**Figure 6.** Duration of the lake ice cover and the timing of calving events at the lake terminus. The size of the squares depict the relative magnitude of events.

position change between winter and summer positions is 280 m. The largest differences occur in the years 2014–2016 with 350, 400 and 410 m difference between minimum and maximum positions.

Ice flow velocities at the glacier front ranged between 1.4 and 2.1 km yr$^{-1}$, with the lowest annual average velocities in 2024 (1.5 km yr) and the highest during 2019–2021 ($\sim$ 1.9 km yr$^{-1}$) (Fig. 7). During these peak years, velocities continued to accelerate even after the runoff season, reaching a maximum in December. As a result, the highest monthly velocities over the full period were on average during October through December. However, no clear seasonal signal can be identified.

Eight kilometres upstream, the ice flow velocities are notably lower, ranging from 0.7 to 0.9 km yr$^{-1}$. Similarly to the
terminus, the highest annual velocities occurred in 2019–2022 and the lowest in 2024. Although velocities are on average highest in May, June, and July, seasonal variability is low, less than 4%. In the upstream velocities, we see a strong acceleration each year coinciding with the beginning of the runoff period.

To assess potential drivers of terminus position variation, we compared monthly retreat rates with air temperature, runoff, and ocean temperature. The strongest correlation was found with runoff (r = 0.6), followed by air temperature (r = 0.5), while
ocean temperature showed a weaker relationship (r = 0.37). As shown in Fig. 8, the runoff period aligns closely with the retreat period, whereas the seasonal ocean temperature cycle lags behind.

## 5   Discussion

In this section, we first discuss the historical evolution and the dynamics of the lake- and marine-terminating glacier before highlighting key differences between the two environments in a broader sense.



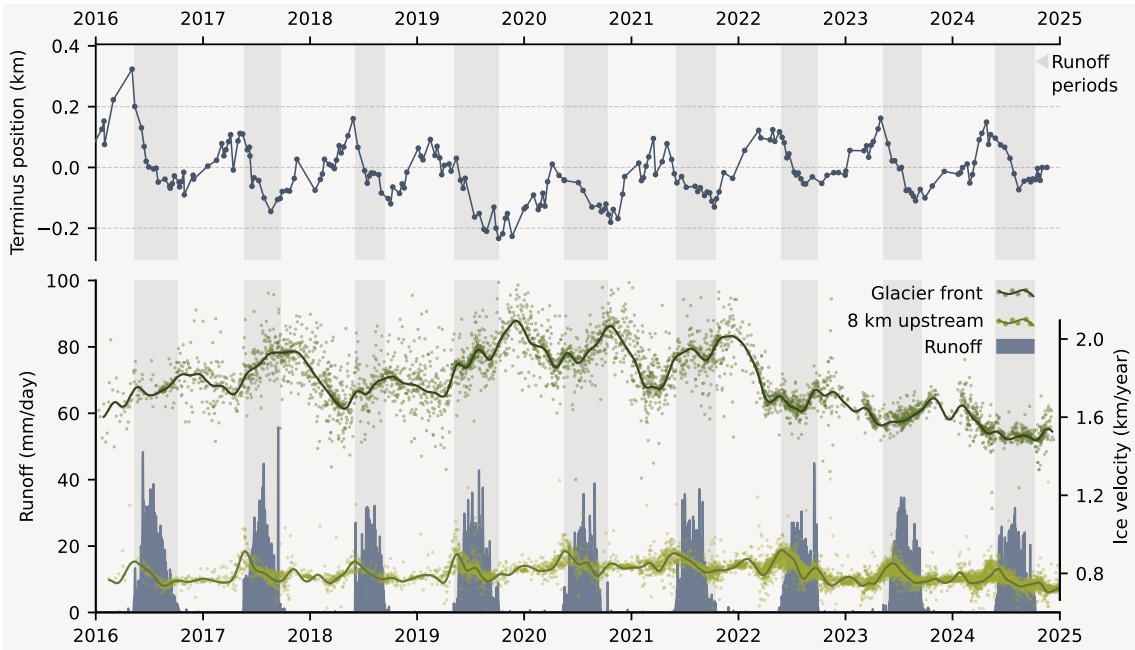

**Figure 7.** Dynamics at the marine terminus. The upper part of the figure shows the seasonal cycle of the glacier front positions relative to the last observation. The lower part shows ice flow velocities at the glacier front and 8 km upstream from ITS_LIVE and TICOI (Gardner et al., 2025; Charrier et al., 2025). The blue bars show daily runoff from RACMO2.3p2 (Noël et al., 2018).

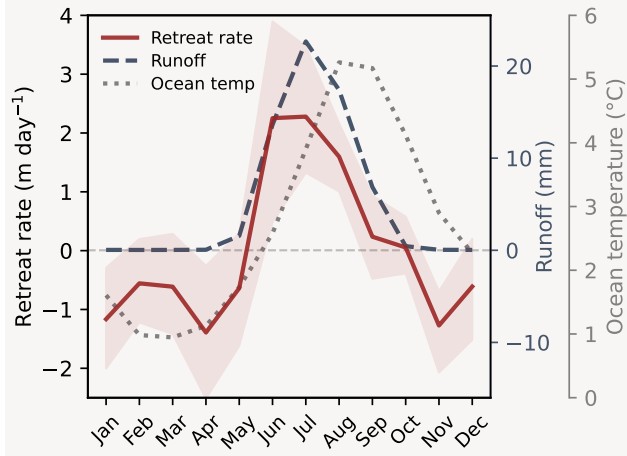

**Figure 8.** Average monthly retreat rates between 2014 and 2024 (red line with confidence interval as shading), average daily runoff per month (blue stippled line) and average monthly ocean potential temperatures (gray dotted line).





## 5.1 The floating ice tongue of the lake-terminating glacier

The flat morphology of the glacier surface, the production of large tabular icebergs, as well as the buoyancy compensated thinning rates suggest that prior to 2012, a floating ice tongue was present at the lake terminus (Fig. 2 and 3). Although we are unaware of other descriptions of floating ice tongues in Greenlandic lakes, they are a common phenomenon in other regions like Patagonia (Warren et al., 2001), Alaska (Boyce et al., 2007; Trüssel et al., 2013), and Iceland (Benn et al., 2007). In Greenland, floating ice tongues are found at marine-terminating glaciers, but only in the very north (Hill et al., 2018; Millan et al., 2023; Wekerle et al., 2024) following the disintegration of several ice tongues further south (Millan et al., 2023; Mouginot et al., 2015; Johnson et al., 2004). Therefore, the presence of the floating ice tongue in Lake Motzfeldt for an extended period (at least 1987–2012), despite being located in the very south of Greenland, can only be explained by the different environment at the terminus. Truffer and Motyka (2016) suggest that floating ice tongues in lakes can occur in more temperate regions due to colder water temperatures and the lack of salinity-driven circulation, which strongly limits subaqueous melt. And indeed, our CTD and temperature measurements in Lake Motzfeldt show a very cold lake, with a depth averaged temperature at the glacier front of < 0.7 °C in August 2025. The lake can sustain such a cold temperature throughout the year due to meltwater input and the immense amount of icebergs present. Any additional energy input through solar radiation or heat advection is likely transferred into the icebergs, efficiently cooling the lake.

Nonetheless, in 2012 and 2013, several large calving events led to the disintegration and massive loss of more than 3 km of the glacier terminus. Our bathymetry data and the historical DEMs indicate that the disintegration of the floating ice tongue was due to the separation from a pinning point at the glacier front. These pinning points exert important backstress on the glacier, and their loss is known to have destabilising effects (Goldberg et al., 2009; Favier et al., 2012). In 1987, the glacier was well grounded at the glacier front. However, substantial thinning led to the almost complete separation from the the pinning point by 2012 (Fig. 3 and 4). Since the disintegration of the floating ice tongue, the glacier has changed to a mode of long advance phases followed by abrupt, large calving events. These calving events still produce large tabular icebergs, which indicates that a floating tongue develops each time during the advance phase, again pointing to low subaqueous melt rates.

## 5.2 Dynamics at the lake terminus

We find a clear seasonality in calving activity, with all major calving events occurring between mid May and mid October (Fig. 6). Remarkably, without exception, this coincides with the lake ice free period but also with the runoff period. We suggest that the lake ice cover may act in a similar way to an ice mélange at marine-terminating glaciers (e.g., Cassotto et al., 2015; Moon et al., 2015; Wehrlé et al., 2023; Meng et al., 2025), providing a buttressing effect that stabilises the glacier front and reduces calving activity during winter. When the lake becomes ice free, this back pressure is removed, allowing existing crevasses and fractures to propagate and ultimately lead to calving events. A comparable relationship has been reported by Mallalieu et al. (2020), who found that periods of lake ice cover limit calving activity at Russell Glacier (Southwest Greenland).

Furthermore, all calving events coincide with the runoff season (Fig. 5). A possible explanation is that runoff, as a consequence of surface melt, can increase buoyant forces through the combined effect of thinning of the ice and an increase in lake



level. Buoyant forces have been identified as important triggers for calving in lacustrine settings, as they lead to the bending of the glacier and promote fracture propagation (Howarth and Price, 1969; Holdsworth, 1973; Warren et al., 2001). These studies

describe several indicators of buoyancy-driven calving, including the presence of large tabular icebergs, melt notches above the waterline, icebergs taller than the glacier front, and fluctuations in lake level. Despite their large magnitude, calving events do not appear to have an effect on the glaciers flow velocities, as they only seem to follow a pattern of a swift increase at the onset of the runoff season and a progressive decrease at the end of the season (Fig. 5).

### 5.3   Historical evolution of the marine terminus

The historical evolution, the seasonal front position, as well as the ice flow velocity largely resemble those of other marine-terminating glaciers in Greenland, with differences discussed below. Dissimilar is the net advance during the pre-satellite era (here 1953–1992) of about 640 m. Notably, Warren and Glasser (1992) describe a retreat of the glacier front between 1942, 1953 and 1981, indicating that the advance must have occurred only between 1981 and 1992, surpassing the initial retreat. Although mean annual air temperatures were considerably lower than average for a few years in the early 1980s, it is difficult

to pinpoint the cause of the advance. For the neighbouring Eqalorutsit Kangilliit Sermiat Weidick (2009) also describes an anomalous advance (during a more extended period 1942–2000), noting a complicated precipitation pattern with large spatial variations over a small area as a possible cause.

After reaching its most advanced position in 1994 the glacier recedes about 1.3 km until 2004, where it remains stable until today. Although the cause of the retreat of other marine-terminating glaciers in Greenland during that period is often induced

by warming ocean temperatures (Wood et al., 2021), we do not have a clear indication of this being the case here. In contrast, the low correlation between front position and ocean reanalysis data (Fig. 8), as well as the presence of a large submarine moraine at the fjord mouth (Weidick, 1959; Vorndran and Sommerhoff, 1974) suggests that the fjord is shielded from warmer bottom waters. Instead, it appears that the position before 2004 was an overextension of the glacier to an unstable position without any visible pinning points. An indication of this is that the glacier front did not remain at a specific position before

2004 but rather changed every year.

### 5.4   Dynamics at the marine terminus

Since 2004 and despite substantial thinning, the glacier oscillates at approximately the same position, which is characterised by a visible narrowing of the fjord through a bedrock protrusion on either side. Seasonal advance and retreat are well-documented patterns for marine-terminating glaciers in Greenland and have a variety of influencing factors, such as fjord and glacier

geometry, air temperature, runoff, ocean temperature, ice velocity, sea ice, and ice mélange (e.g., Schild and Hamilton, 2013; Moon et al., 2015; Black and Joughin, 2023; Greene et al., 2024). In this study, we found that the seasonal front position of Qooqqup Sermia is best explained by surface runoff (out of runoff, air and ocean temperature). Similarly to our findings, (Fried et al., 2018) conclude that, in general, runoff is the best predictor for grounded glaciers with a small (< 500 m) seasonal amplitude, which Qooqqup Sermia also classifies as. Runoff, which enters the fjord as subglacial discharge, can lead to the

formation of buoyant plumes along the glacier front (Hewitt, 2020). These plumes entrain ambient, warmer water and can





efficiently melt the glacier front. Besides the direct melting of the glacier front, plumes can also have an undercutting effect, which can trigger calving events (Rignot et al., 2015; Fried et al., 2015; Slater et al., 2017). During the time of fieldwork, as well as in satellite images, a plume was observed at the front of Qooqqup Sermia. In combination with the fact that the glacier front position is best explained by runoff, this indicates that frontal ablation at the marine terminus is dominated by submarine
melt-induced calving events.

As is typical for calving glaciers, Qooqqup Sermia's flow velocity also increases towards the terminus, thereby more than doubling its speed from 8 km upstream to the glacier front. The upstream ice-flow velocities remain relatively constant throughout the observation period, with a notable spike at the beginning of the runoff season each year. This short-term increase in flow velocities is also observed in other regions of Greenland (van de Wal et al., 2008) and is linked to the efficiency of subglacial
drainage when meltwater or rainwater enters the system (Bartholomaus et al., 2008; Schoof, 2010). Although in most years, the ice flow velocities close to the glacier front also increase at the beginning of the runoff season, they show an anomalous speedup throughout and even after the runoff season in some years (Fig. 7). This signal is not visible in the upstream velocities, suggesting that it originates at the glacier front. Interestingly, the highest flow velocities coincide with the most retreated glacier positions. This could indicate that a far retreated position of the glacier provides less back stress to ice masses upstream from
there, resulting in increased flow velocities.

### 5.5    Key differences between the marine and lake environment

In the previous chapters, we illustrated that the lake- and marine-terminating glacier exhibit contrasting dynamics, despite being subject to the same climatic conditions. Following that, we now analyse the main differences between the two environments more broadly. Generally, lake environments differ from marine environments for two inherent reasons:

1. A lake is a semi-closed system where no transport of large quantities of warm water masses toward the glacier front from distant sources takes place. The energy input to the lake is limited to long and short wave radiation, heat exchange with the atmosphere, and advected heat (e.g., rainwater, streams, and groundwater) (Wetzel and Likens, 2000). In the fjord, on the other hand, warm water masses can be transported to the glacier front through ocean circulation and exchange with more distant water masses. The intrusion of warm Atlantic waters deep into fjords is a commonly observed phenomenon
for tidewater glaciers in all regions of Greenland (e.g., Holland et al., 2008; Rignot et al., 2012; Wood et al., 2021; Wekerle et al., 2024).

2. Lakes generally consist of freshwater and therefore lack any salinity driven circulation. At marine-terminating glaciers, however, salinity driven circulation, and specifically the formation of meltwater plumes at the glacier front, have a strong impact on the glacier by increasing submarine melt rates (Jenkins, 2011), promoting undercutting, and consequently
calving (Slater et al., 2017).

Both of the above differences result in lower subaqueous melt rates in lakes compared to the fjord. The low melt rates, in turn, can be the cause of different dynamics at the calving front, as they allow for the formation of a floating ice tongue and limit calving due to undercutting. Therefore, we argue that subaqueous melt is the main process for the contrasting dynamics



between the two environments. While we have indications that subaqueous melt rates are indeed very low at lake Motzfeldt
and other lake-terminating glaciers (Trüssel et al., 2013; Truffer and Motyka, 2016), direct or indirect observations are missing,
alongside conceptual studies investigating the possibility of plumes in lacustrine settings.

Besides the aforementioned differences, marine terminating glaciers are often subject to larger tidal variations, which can
trigger calving through a change in the stress field (Bartholomaus et al., 2015; Holmes et al., 2023; Marsh et al., 2025). Lakes
can also have substantial water level variations (Dømgaard et al., 2024), as we also show in this study. However, these events are
often less frequent within a year compared to tides. Dømgaard et al. (2024) also found that a large number of lakes periodically
drain in the form of glacial lake outburst floods. However, relatively little is known about the ice-dynamical effects of draining
lakes on glaciers.

## 6   Conclusions

In this study, we compared two adjacent glacier termini of the Qooquup Sermia glacier system, one of which is lake-terminating
and the other is marine-terminating. From our analysis, we draw the following conclusions:

- The marine terminus of the Qooqqup Sermia system is fast flowing, grounded and characterised by rapid, melt-driven
  calving. The lake terminus, on the other hand, develops a floating ice tongue during prolonged advance phases and
  experiences large buoyancy-driven calving events.

- Despite being subject to the same climatic forcing and fed by the same upstream ice masses, the two termini show
  contrasting front dynamics in the long- and short-term. These results stress that lake- and marine-terminating glaciers
  require different parameterisations for front dynamics when included in model simulations of the Greenland Ice Sheet.

- We argue that low subaqueous melt rates in lakes are the main driver of differences at our study site and in general. Here,
  low subaqueous melt rates led to the development of a floating ice tongue, which consequently changes the calving
  behaviour. However, direct and indirect observations of subaqueous melt rates in lakes are lacking and further research
  is needed.

- We provide evidence that lake ice plays an important role in limiting calving activity. However, a larger scale study
  would be desirable to confirm this.

- The massive retreat of more than 3 km of the lake terminus within one year highlights the possibility of rapid mass loss
  at lake-terminating glaciers in Greenland, stressing their importance for the future evolution of the Greenland ice sheet.




**Appendix A**

## A1  Temperature and pressure time series

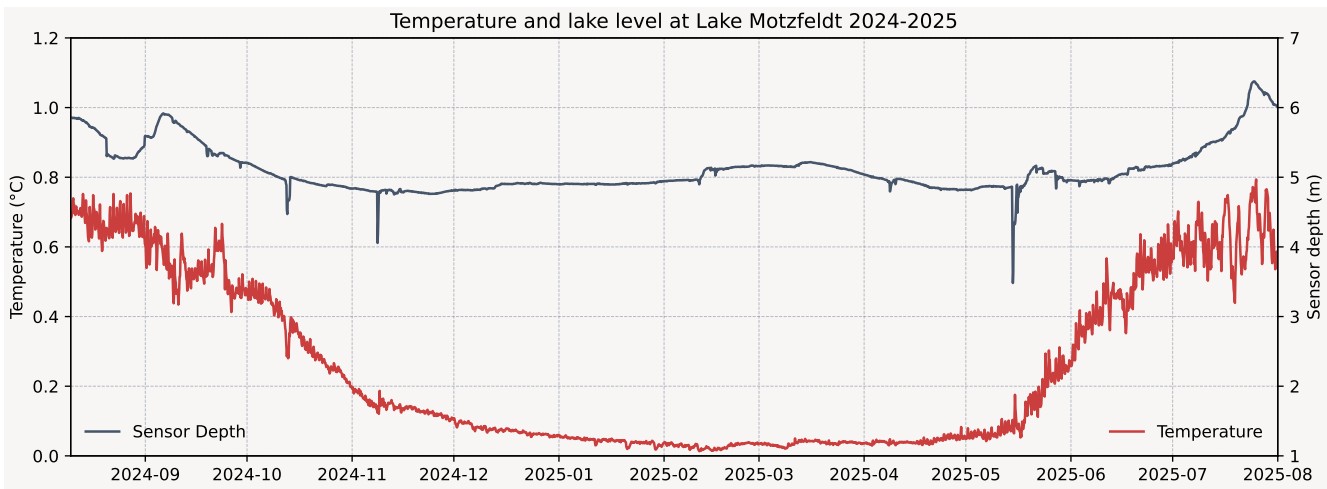

**Figure A1.** Temperature and sensor depth (meter water pressure) time series of the RBR sensor in Lake Motzfeldt. The location of the sensor is shown in Fig. 1. The sharp spikes in the depth data (pressure decrease) are due to the rope to which the sensor was attached, being moved, likely by icebergs or drifting lake ice. A pressure increase during winter (e.g mid February) can be linked to snow accumulating on the lake ice.

## A2  CTD measurements

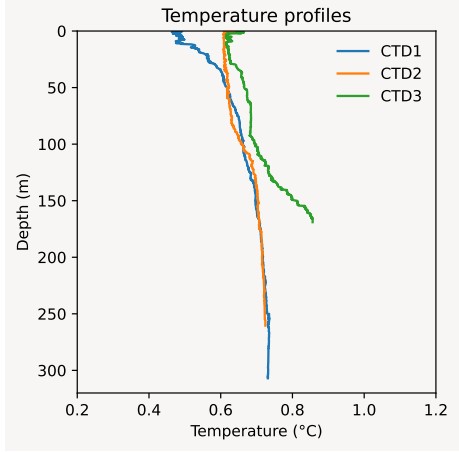

**Figure A2.** CTD measurements in Lake Motzfeldt. The locations are indicated in Fig. 1



*Code and data availability.* Data collected for this publication was made publicly available at the following links: Lake bathymetry at https://doi.org/10.5281/zenodo.17600758 (Vacek et al., 2025b), lake temperature and pressure data at https://doi.org/10.5281/zenodo.17601786
(Vacek et al., 2025a), all other supporting data at https://doi.org/10.5281/zenodo.17602738 (Vacek, 2025b), and figure production code at https://doi.org/10.5281/zenodo.17603519 (Vacek, 2025a).

Further data used in this study can be found here: ArcticDEM at https://www.pgc.umn.edu/data/arcticdem/, ITS_LIVE ice velocity data at https://its-live.jpl.nasa.gov/, ORAS5 Ocean reanalysis data at https://cds.climate.copernicus.eu/datasets/reanalysis-oras5, and Narsarsuaq air temperature data from the Danish Meteorological Institute (DMI) at https://www.dmi.dk/publikationer.

*Author contributions.* FV, FN, WI and RW were involved in conceptualisation. FV, FN, MZ and DB collected field data. FV curated the data, conducted the analysis, produced figures and the manuscript draft. All authors were involved in assessment of the results and contributed to the reviewing and editing of the manuscript.

*Competing interests.* The authors declare that they have no conflict of interest.

*Acknowledgements.* We thank Marcel van Maarseveen for invaluable support leading up to and during the fieldwork. We are grateful to
Andreas Vieli for lending us instruments and equipment that were crucial for the success of the measurements. Furthermore, we thank Greenland Guidance and the Blue ice team for their excellent support in Narsarsuaq.



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
