# Peer review of "Contrasting dynamics of lake- and marine-terminating glaciers under same climatic conditions"

_EGUsphere, 2025_

## Referee Comment (RC1)

This study presents a well-designed comparison between two branches of the same glacier system—one marine-terminating and the other lake-terminating—to investigate how glaciers with different terminus types respond dynamically under identical climatic conditions. The choice of study site is particularly thoughtful: because both branches originate from a common ice divide, they naturally experience the same regional climate, allowing for a robust controlled comparison. The authors support their analysis with comprehensive datasets, including multi-source remote sensing observations and valuable field measurements from two campaigns. The manuscript is clearly written, and the figures are well-designed and effectively illustrate the key findings. Overall, I recommend this paper for publication after minor revisions.

However, I note three specific points where my interpretation differs from the authors'. These points are offered to refine the discussion and enhance the manuscript's analytical precision:

(1) **Terminus-driven dynamics of the marine-terminating branch:**

The velocity variations of the marine-terminating branch appear to be primarily terminus-driven rather than runoff-driven. Specifically, the annual onset of acceleration aligns more closely with the start of terminus retreat, typically preceding the melt season, and the deceleration coincides with the end of retreat, often extending beyond the runoff period (as noted by the authors in Line 363). These patterns strongly suggest that terminus position and calving dynamics, not surface runoff, dominate the glacier's flow variability. Such behavior is well-documented for many Greenlandic outlet glaciers (e.g., Moon et al., 2014; Vijay et al., 2019, 2021) and should not be considered "anomalous" (Line 361). That said, short-term velocity pulses during the melt season (e.g., in 2019–2021) may indeed reflect runoff-related processes. Recent work on Eqip Sermia (Zhang et al., 2025), we observed similar characteristics: seasonal acceleration extending beyond the melt period, tight coupling between retreat and speed-up timing, and velocity pulses during melt seasons. This observation may provide useful context for interpreting this system.

(2) **Formation of the floating ice tongue at the lake-terminating branch:**

I agree with the authors that the relatively flat surface profile of the glacier suggests a floating ice tongue may have been present at the lake terminus prior to 2012. However, I am not fully convinced that "low subaqueous melt rates" were the primary driver of its formation. Instead, I

believe the glacier geometry, particularly the exceptional depth of Lake Motzfeldt (Line 220), played the decisive role. A floating terminus arises when ice thins sufficiently to reach buoyancy equilibrium over deep water; subaqueous melt rates may modulate stability and retreat, but it is unclear how subaqueous melt rates alone could induce flotation. To put it another way: if another lake-terminating glacier terminates in a shallow basin, I believe it is unlikely to develop a floating tongue even under equally low subaqueous melt rates.

(3) **Calving seasonality and floating-glacier behavior:**

The authors note that the lake-terminating glacier exhibits long advance phases punctuated by abrupt, large calving events. However, this floating-glacier behavior is not unique to northern glaciers in Greenland; similar patterns are observed at several marine-terminating glaciers in central Greenland, such as Helheim, Kangerdlussuaq, and the northern branch of Rink Glacier. What appears distinctive about the lake-terminating glacier in this study is that its major calving events consistently occur during the melt season, whereas the aforementioned marine glaciers experience large calving events throughout the year. This seasonal contrast may be linked to the lake's low background subaqueous melt rates: only during the melt season does subaqueous melt increase sufficiently to destabilize the terminus and trigger large-scale calving. While this interpretation remains speculative, it warrants brief discussion, as it could point to a fundamental mechanistic difference between lacustrine and marine floating termini.

**Specific comments:**

Figure 5: Legend for large calving event is missing.

**Reference**

Moon, T., Joughin, I., Smith, B., van den Broeke, M. R., van de Berg, W. J., Noël, B., and Usher, M.: Distinct patterns of seasonal Greenland glacier velocity, Geophysical Research Letters, 41, 7209–7216, https://doi.org/10.1002/2014GL061836, 2014.

Vijay, S., Khan, S. A., Kusk, A., Solgaard, A. M., Moon, T., and Bjørk, A. A.: Resolving Seasonal Ice Velocity of 45 Greenlandic Glaciers With Very High Temporal Details, Geophysical Research Letters, 46, 1485–1495, https://doi.org/10.1029/2018GL081503, 2019.

Vijay, S., King, M. D., Howat, I. M., Solgaard, A. M., Khan, S. A., and Noël, B.: Greenland ice-sheet wide glacier classification based on two distinct seasonal ice velocity behaviors, Journal of Glaciology, 67, 1241–1248, https://doi.org/10.1017/jog.2021.89, 2021.

Zhang, E., Catania, G., Smith, B., Felikson, D., Csatho, B., and Trugman, D. T.: Compounding sub-seasonal variations in Greenland outlet glacier dynamics revealed by high-resolution observations, EGUsphere [preprint], https://doi.org/10.5194/egusphere-2025-4216, 2025.